# A streamlined platform for analyzing tera-scale DDA and DIA mass spectrometry data enables highly sensitive immunopeptidomics

Lei Xin[1,3], Rui Qiao[1,3], Xin Chen[1,3], Hieu Tran [2,3], Shengying Pan[1], Sahar Rabinoviz[1], Haibo Bian[1], Xianliang He[1], Brenton Morse[1], Baozhen Shan[1✉] & Ming Li [2✉]

Integrating data-dependent acquisition (DDA) and data-independent acquisition (DIA) approaches can enable highly sensitive mass spectrometry, especially for imunnopeptidomics applications. Here we report a streamlined platform for both DDA and DIA data analysis. The platform integrates deep learning-based solutions of spectral library search, database search, and de novo sequencing under a unified framework, which not only boosts the sensitivity but also accurately controls the specificity of peptide identification. Our platform identifies 5-30% more peptide precursors than other state-of-the-art systems on multiple benchmark datasets. When evaluated on immunopeptidomics datasets, we identify 1.7-4.1 and 1.4-2.2 times more peptides from DDA and DIA data, respectively, than previously reported results. We also discover six T-cell epitopes from SARS-CoV-2 immunopeptidome that might represent potential targets for COVID-19 vaccine development. The platform supports data formats from all major instruments and is implemented with the distributed high-performance computing technology, allowing analysis of tera-scale datasets of thousands of samples for clinical applications.

[1] Bioinformatics Solutions Inc., Waterloo, Ontario, Canada. [2] David R. Cheriton School of Computer Science, University of Waterloo, Waterloo, Ontario, Canada. [3] These authors contributed equally: Lei Xin, Rui Qiao, Xin Chen, Hieu Tran. ✉email: bshan@bioinfor.com; mli@uwaterloo.ca

Throughout the three decades-long history of mass spectrometry (MS)-based proteomics, improving the sensitivity of peptide and protein identification has always been one of the most important research objectives[1,2]. Numerous efforts have been made to increase the detection power in different stages of MS experiments, including sample preparation, instrumentation, data acquisition strategy, and data analysis. For instance, data-independent acquisition (DIA) strategy allows the fragmentation of all precursor ions within a certain range of m/z and retention time, thus producing a complete record of all peptides in a sample[3,4]. This is in contrast to data-dependent acquisition (DDA) strategy which only selects a few most intense precursor ions for fragmentation. Similarly, different analysis approaches and software tools have been developed for peptide and protein identification from MS data, including de novo peptide sequencing, protein database search, and spectral library search[5–13]. The sensitivity is especially critical for immunopeptidomics applications due to the difficulty of obtaining sufficient amount of samples and the complexity of human leukocyte antigen (HLA) peptidomes in cancer or virus-infected cells[14–16]. Furthermore, recent studies of immunopeptidomics-based vaccines against cancer[17,18] or infectious diseases such as COVID-19[19,20] have shown that only a few T-cell epitopes among several thousands of identified HLA peptides represent effective targets for vaccine development. Thus, the sensitivity needs to be high enough to catch all of those relevant T-cell epitopes, otherwise identifying thousands of HLA peptides is not really meaningful.

While continuous efforts are being made to improve each component of MS data acquisition and analysis, recent studies[21,22] have suggested that integrating different MS data acquisition strategies and analysis approaches might significantly improve the sensitivity of MS-based immunopeptidomics. However, simply combining different tools or resources may not be an optimal solution because (i) they were originally developed with different designs and principles and (ii) they are difficult to scale up together to simultaneously analyze thousands of datasets for clinical applications.

Here we propose a platform, PEAKS Online, that seamlessly integrates two types of data acquisition, DDA and DIA, and three data analysis approaches, including de novo peptide sequencing, protein database search, and spectral library search. The platform is designed to streamline the analysis of DDA and DIA data by applying consistent algorithms, confidence score calculation, false discovery rate (FDR) estimation, and visualization across the board. Moreover, a deep learning-based approach is applied throughout the platform, from basic tasks such as spectrum or retention time predictions, to complicated processes such as de novo sequencing, database or spectral library search. The platform is also implemented with the latest distributed high-performance computing technology, allowing high-throughput analysis of tera-scale datasets of thousands of samples for clinical applications on modern cloud computing platforms. We evaluated our platform on multiple benchmark datasets and identified 5–30% more peptide precursors than current state-of-the-art systems. When evaluated on three DDA immunopeptidomics datasets, our standard database search identified 1.7–4.1 times more peptides, while our rescoring with deep learning-predicted spectra identified 1.0–1.4 times more peptides than previously reported results. In another evaluation on a DIA immunopeptidomics dataset, our integrated workflow of spectral library search, database search, and de novo sequencing together identified 1.4–2.2 times more peptides than previously reported results. We also applied our platform to the SARS-CoV-2 HLA immunopeptidome and discovered six T-cell epitopes from infected cells that may represent potential targets for COVID-19 vaccine development.

## Results

**An ultra-sensitive and streamlined platform for DDA and DIA mass spectrometry.** Our PEAKS Online platform, an example integrated workflow for DIA analysis on the platform, and the performance evaluations are described in Fig. 1. The platform integrates three main computational approaches, including spectral library search, database search, and de novo sequencing for the analysis of both DDA and DIA data. Each of the three computational approaches can be performed separately or they can be used together in a workflow. Figure 1b describes an example workflow where all three computational approaches are consecutively performed on a DIA dataset to achieve the highest possible sensitivity. The list of peptides identified by the three computational approaches are then used to build a spectral library, and a final search of the whole dataset is performed against this new library. The final search is meant to re-confirm the identified peptides and to provide a unified global FDR.

The integration is streamlined under a unified framework that (i) applies consistent analysis algorithms to both DDA and DIA data; (ii) systematically controls the FDR of peptides identified from all three computational approaches; and (iii) follows a deep learning-based and data-driven principle. For instance, a feature-based principle is applied across the three computational approaches and to both DDA and DIA data: precursor features are first detected from LC-MS map; MS/MS spectra are then grouped with their corresponding precursors based on their co-elution profiles; each precursor and its associated MS/MS spectra are then fed to any of the three computational approaches to identify the peptide[23]. The FDR of identified peptide-spectrum matches (PSMs) is calculated based on a consistent target-decoy approach, where random decoy peptides and spectra are generated by randomly permuting the peptide sequences and the fragment ions, respectively. In the case of de novo sequencing, de novo peptides can be put into a new database and an additional database search is performed to estimate the FDR of de novo peptides. Deep learning applications are employed at multiple stages throughout the platform, including MS1 isotope feature detection[24,25], de novo sequencing for both DDA and DIA[23,26,27], spectrum, retention time, and collision cross section predictions[28–30] (Supplementary Figs. 1, 2, Supplementary Table 1). Deep transfer learning is also used to refine a public spectral library to adapt to a specific MS instrument setting. More details of our platform are described in the "Methods".

We evaluated PEAKS Online on three DIA benchmark datasets from recent studies, including Muntel et al.[31], Xuan et al.[32], and Association of Biomolecular Resource Facilities (ABRF) study[33]. Muntel et al. attempted to establish an optimal MS instrumentation and data analysis strategy to identify 10,000 proteins in one single DIA run. Xuan et al. tried to standardize a robust, sensitive, and reproducible workflow for DIA data generation and analysis across eleven international laboratories of the international Cancer Moonshot consortium[34]. We compared our platform to two other state-of-the-art software suites for DIA data analysis, DIA-NN[35] and Spectronaut[13]. In this evaluation, all three tools were run using their spectral library search option. As shown in Fig. 1c, d, our platform identified 5–30% more peptide precursors than DIA-NN and Spectronaut at 1% FDR on both benchmark datasets. In addition, our test on the ABRF benchmark dataset demonstrated that our platform produced consistent identification results across different DIA runs (Venn diagram in Fig. 1e).

We also evaluated PEAKS Online on three DDA immunopeptidomics datasets, including two HLA monoallelic cell lines from Sarkizova et al.[36] and one native melanoma sample from Bassani-Sternberg et al.[14]. Figure 1f shows that, in standard database search mode, PEAKS Online outperformed MaxQuant[8] by 1.7–4.1 times.

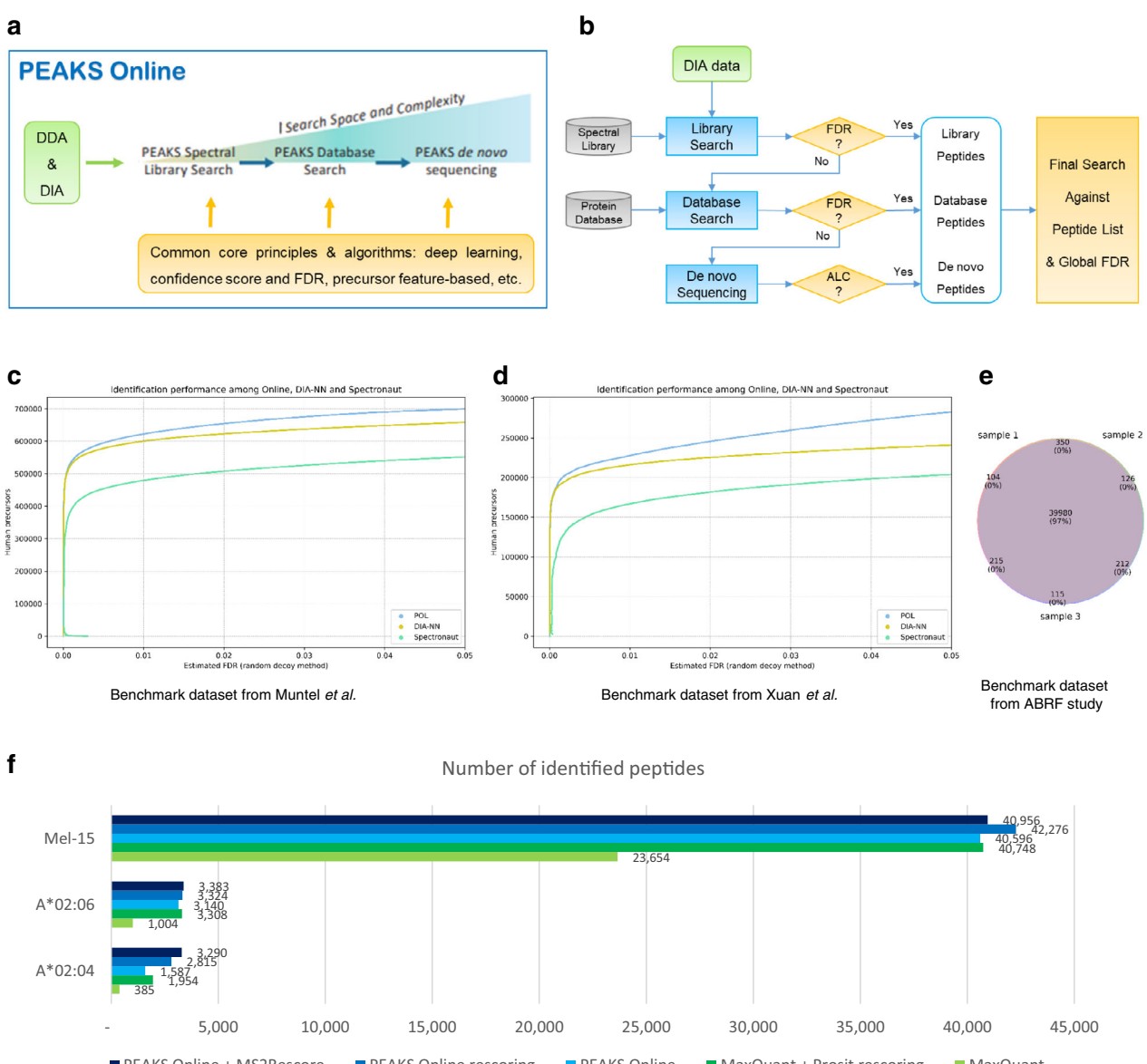

**Fig. 1 PEAKS Online: an ultra-sensitive and streamlined platform for DDA and DIA mass spectrometry. a** An overview of data acquisition strategies, analysis approaches, and core algorithms of the platform. **b** An example integrated workflow for DIA data analysis on the platform. **c–e** Performance evaluation of PEAKS Online, DIA-NN, and Spectronaut on three DIA benchmark datasets: Muntel et al.[31], Xuan et al.[32], and ABRF study[33]. **f** Performance evaluation of PEAKS Online, MaxQuant, Prosit, and MS2Rescore on three DDA immunopeptidomics datasets: Mel-15 from Bassani-Sternberg et al.[14]; A*02:04 and A*02:06 from Sarkizova et al.[36]. (DDA: Data-Dependent Acquisition; DIA: Data-Independent Acquisition; FDR: False Discovery Rate; ALC: Average of the Local Confidence score; POL: PEAKS Online; ABRF: Association of Biomolecular Resource Facilities).

Last but not least, PEAKS Online is designed to achieve high parallelism, high scalability, and high stability with the modern distributed high-performance computing technology (Supplementary Fig. 3). The platform is built on top of the Java Akka Toolkit. We implemented a master-workers computation cluster with Akka's actor based messaging model. Following the map-reduce pattern, we split larger projects into fractions, run them in parallel on different worker nodes, and aggregate the results in the end to present to the user. A distributed data storage, Apache Cassandra, is integrated in our system for data persistence. So datasets produced by different worker nodes can be shared and accessed, which allows us to hand over tasks between workers and perform data aggregation in the end. More details of PEAKS Online's architecture can be found in the Methods.

Supplementary Table 2 demonstrates the high-performance computing capability of our platform to perform high-throughput analysis on a large project of more than four thousands samples with 3.7 terabytes of data. The analysis was performed on the Amazon Web Services cloud system with 512 CPU cores and 1 TB RAM, and was completed within just less than 20% of the time required for the MS experiments. Another example in Supplementary Fig. 4 demonstrates the scalability of PEAKS Online, i.e., how the computational time efficiently decreased with the increasing number of CPU cores.

**Applications in cancer immunopeptidomics**. MS-based immunopeptidomics has significant clinical impacts in the development of vaccines against cancer or infectious diseases. Mass spectrometry provides direct evidence of HLA peptides displayed on the

surface of cancer or virus-infected cells, which represent potential targets of vaccines. However, the presentation pathway of HLA peptides, including their cleavage, is not well-understood, as opposed to traditional tryptic peptides. Hence, it is important to assess the ability of a search engine to identify HLA peptides from DDA and DIA data.

As mentioned earlier, Fig. 1f shows the performance evaluation of PEAKS Online and MaxQuant on three DDA immunopeptidomics datasets from refs. [36] and [14]. It is important to note that, since most search engines were not originally designed for HLA peptides, the application of deep learning-predicted spectra to rescore database search results can substantially improve the number of identified peptides[22,37]. For instance, Wilhelm et al.[22] recently proposed Prosit, a deep learning model that was trained specifically on synthetic HLA peptides to boost the identification. We show in Fig. 1f that both standard PEAKS Online and its rescoring results consistently outperformed the results of MaxQuant and MaxQuant plus Prosit rescoring by 1.7–4.1 and 1.0–1.4 times, respectively, across three datasets. We also compared PEAKS Online rescoring results to MS²Rescore[37] and found that they were comparable for two datasets Mel-15 and A*02:06, whereas for dataset A*02:04 MS²Rescore identified 17% more peptides. More details about our rescoring procedure can be found in the Methods.

Next, we evaluated the performance of PEAKS Online on a DIA immunopeptidomics dataset published recently by Pak et al.[21]. The dataset includes both DDA and DIA measurements of the immunopeptidome sample from the human B cell line RA957. In their study, Pak et al. used MaxQuant[8] to analyze DDA data to build sample-specific spectral libraries and then used Spectronaut[13] to perform spectral library search on DIA data. We performed the same analysis workflow using our platform and achieved substantially higher sensitivity without compromising accuracy.

Figure 2a shows that we identified a total of 26,558 peptides from three DDA runs, which was 79.6% higher than previously reported by Pak et al. (14,789 peptides). As a result, our spectral library had much better coverage and enabled more sensitive DIA search. Our spectral library search identified a total of 24,290 peptides from two DIA runs, which is 66.4% higher than previously reported (14,600 peptides). Figure 2b shows the Venn diagram of our identified peptides, the results from Pak et al., and the HLA-I peptides from the Immune Epitope Database (IEDB)[16]. There were 10,601 new peptides identified by our spectral library search but not reported by Pak et al., and among them, 3809 peptides can be found in the IEDB. Further binding motif deconvolution analysis by MixMHCp 2.1[38] shows that the peptides reported by Pak et al. and by our spectral library search were clustered into the same set of six binding motifs and with similar proportions across those motifs (Fig. 2c). We also investigated the spectra for which our spectral library search and Pak et al. identified different peptide sequences (Supplementary Data 1). We found 241 such spectra, and for 195 (80.9%) of them, the peptides identified by our spectral library search had stronger binding affinities than the ones by Pak et al., as predicted by NetMHCpan[39]. An example PSM comparison is shown in Supplementary Fig. 5, where we also used an independent spectrum prediction tool MS²PIP[40] to validate the identified peptides. Both spectrum and binding prediction results show that our identified peptide was more accurate.

To further increase the sensitivity, we applied the integrated DIA workflow described in Fig. 1b to the sample RA957. In particular, spectral library search, database search, and de novo sequencing were consecutively performed on the DIA data. The list of peptides identified by the three computational approaches were used to build a new spectral library. A final search of the whole dataset was performed against the new library with a unified global FDR of 1%. As shown in Fig. 3a, the database search and de novo sequencing identified 6853 and 925 extra peptides, respectively, on top of the library search results (i.e., about 32.1% improvement). In total, the integrated workflow by PEAKS Online identified 31,984 peptides. When compared to the results reported by Pak et al., we identified 2.2 times more peptides than their sample-specific library approach (14,600 peptides) and 1.4 times more peptides than their multi-HLA BigLib library or the Prosit-predicted library approaches (22,532 peptides)[21].

The lengths of both new and previously reported peptides followed the characteristic length distribution of HLA-I peptides (Fig. 3b). We also found that the distributions of peptide lengths, PSM scores, and retention times (RTs) of the peptides identified by different computational approaches of PEAKS Online were comparable (Fig. 3b–d). In addition, the predicted RTs were highly correlated with the experimental RTs. Since this DIA dataset RA957 is HLA data, we also ran NetMHCpan on all identified peptides by PEAKS Online and found that about 88% of them were predicted as weak binders (rank ≤ 2%), 76% as strong binders (rank ≤ 0.5%). The complete details of all PSMs identified by PEAKS Online are provided in Supplementary Data 2. Furthermore, Supplementary Fig. 6 shows the PSM score distributions of random decoy peptides and target peptides. As expected for FDR control, the score distribution of the random decoy peptides was indeed located at the lower end of the score distribution of the target peptides. We also performed a mass shift test[41], where the precursor masses were shifted by adding 100 Dalton and the database search was repeated with the new precursor masses. As expected, the target score distribution was reduced and nearly identical to the decoy one because the target peptides no longer matched the precursor masses after the shifting.

It's worth mentioning that we applied a very stringent control on the peptides identified by de novo sequencing. First, we only selected peptides with positional confidence scores of at least 90 (in a scale of 0–100), which means that every amino acid was supported by both b- and y- ions in our de novo sequencing algorithm. Second, the selected de novo peptides were further subjected to the final search with a global FDR of 1% as mentioned above. As shown in Fig. 3b–d, the de novo peptides had similar distributions of peptide lengths, PSM scores, and RTs as those identified by spectral library search or database search. There was a slightly elevated number of 8-mers in the length distribution of the de novo peptides. Supplementary Fig. 7 shows the PSMs of some example de novo peptides together with their precursor profiles and fragment ion profiles. To validate those de novo peptides, we also ran an independent spectrum prediction tool MS²PIP[40] and found that the predicted spectra were highly correlated with the experimental spectra (Supplementary Fig. 7d). The amino acids at the anchor positions were consistent with the binding motifs of the sample HLA alleles (Supplementary Fig. 7e).

All of the above results strongly confirmed the validity of the peptides identified by our integrated DIA workflow using PEAKS Online. Thus, our platform considerably boosted the sensitivity of HLA peptide identification while still accurately controlling the specificity. More details of the analysis of cancer immunopeptidomics datasets can be found in Supplementary Note 1.

**Analysis of SARS-CoV-2 HLA-I immunopeptidome from COVID-19 infected cells.** We applied our platform to analyze the HLA-I immunopeptidome of SARS-CoV-2-infected A549 cells and HEK293T cells, which were published recently by ref. [19]. A549 cells are human lung cells that are targeted by SARS-CoV-2,

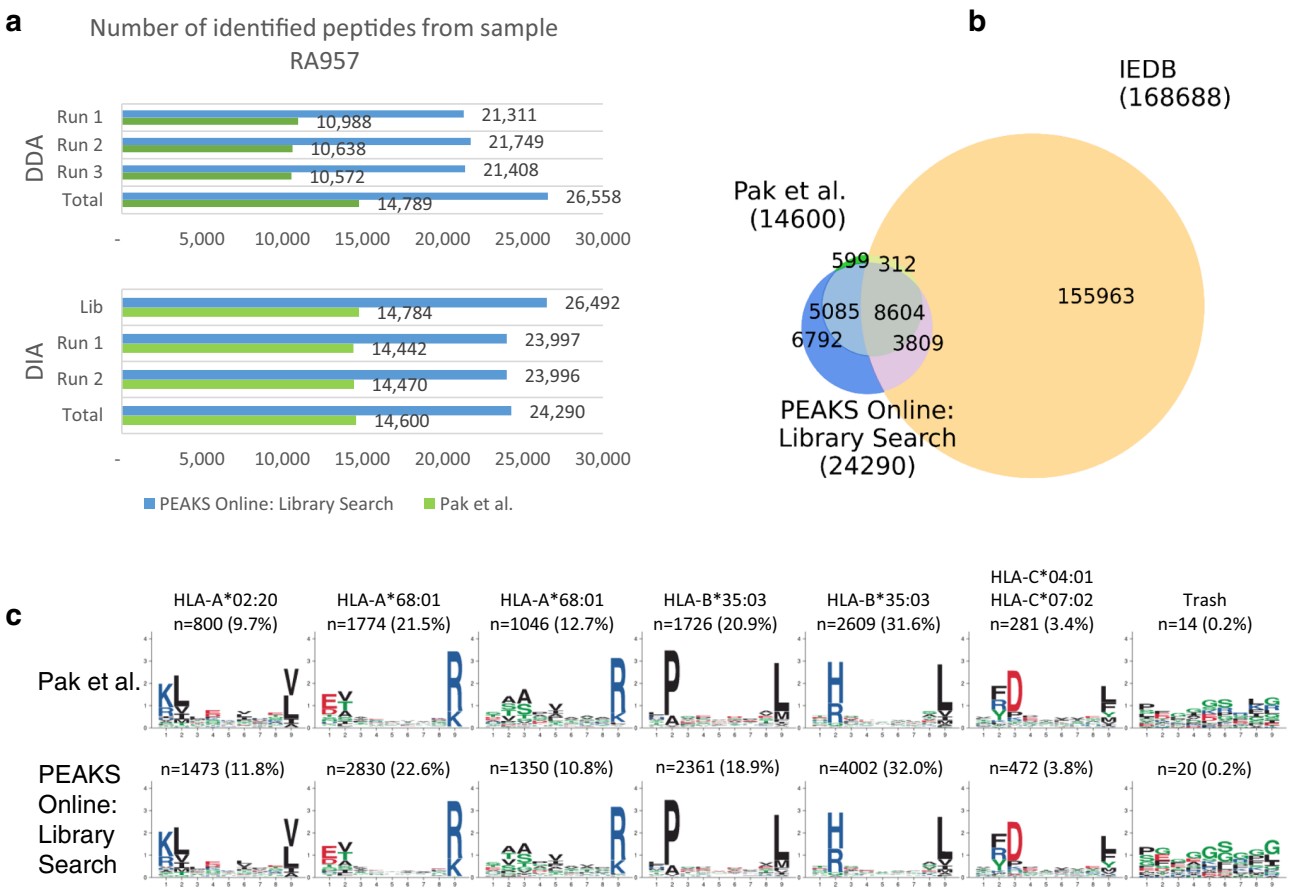

**Fig. 2 Comparison of DIA spectral library search results by Pak et al.[21] and PEAKS Online on the immunopeptidomics sample RA957. a** Sample-specific spectral libraries were built from the database search results of three DDA runs. The libraries were subsequently used to search two DIA runs. **b** Venn diagram of the identified peptides and the HLA-I peptides from the Immune Epitope Database (IEDB). **c** Binding motif deconvolution of the identified peptides by MixMHCp 2.1. (DDA: Data-Dependent Acquisition; DIA: Data-Independent Acquisition; IEDB: Immune Epitope Database).

whereas HEK293T cells express HLA-A*02:01, B*07:02, and C*07:02 alleles that have high coverage in the human population. We identified 6418 and 1932 HLA-I peptides from the infected A549 and HEK293T cells, respectively. Their characteristic length distributions and their binding motifs are presented in Supplementary Fig. 8. Notably, more than 2000 peptides had not been reported in the original study, potentially including new T-cell epitopes for vaccine targets.

We focused on the HLA-I peptides that are derived from the SARS-CoV-2 genome. Supplementary Data 3 lists all SARS-CoV-2 HLA-I peptides that were identified by Spectrum Mill in the original study and by our platform. First, we noted that our platform did not report four peptides that were found by Spectrum Mill. Further investigations showed that these peptides have low confidence scores (from 5.26 to 9.01) assigned by Spectrum Mill. For example, Supplementary Fig. 9 shows that the peptide ELPDEFVVVTV (Spectrum Mill score 9.01) does not have a good peptide-spectrum match and the experimental spectrum is not correlated with the predicted spectrum by Prosit[28], where several fragment ions with high intensities cannot be annotated. Thus, there is not enough evidence to support the representation of these peptides on the cell surface and they may not be reliable targets for vaccine development.

More importantly, we identified six SARS-CoV-2 HLA-I peptides that had not been reported in the original study for the infected A549 cells and HEK293T cells (Fig. 4). Moreover, five of them have been confirmed to activate T-cell response in other studies. For example, the epitope SIIAYTMSL is part of the Spike

glycoprotein of SARS-CoV-2 and has been tested in 13 T-cell assays in 10 different studies[20]. Another epitope HLVDFQVTI is part of the ORF6 protein of SARS-CoV-2 and can also be found in SARS-CoV-1, it has been tested in 13 T-cell assays in 8 different studies. The peptide-spectrum matches supporting the identification of these T-cell epitopes are also provided in Fig. 4.

Thus, the analysis results of SARS-CoV-2 HLA-I immunopeptidome confirm not only the sensitivity but also the specificity of our platform. Both of them are crucial for identifying the correct targets and reducing the number of uncertain candidates for vaccine development, especially when imunnopeptidomics samples are limited and signals are weak and noisy. More details of the analysis of SARS-CoV-2 HLA-I immunopeptidome can be found in Supplementary Note 2.

## Discussion

In this study, we presented PEAKS Online, an ultra-sensitive and streamlined platform for DDA and DIA mass spectrometry analysis. The platform integrates deep learning-based solutions of spectral library search, database search, and de novo sequencing under a unified framework, which not only boosts the sensitivity but also accurately controls the specificity of peptide identification. Indeed, we showed that our platform was able to double the number of identified HLA peptides in cancer immunopeptidomics datasets, while correctly preserving their characteristic binding motifs. Similarly, we identified six T-cell epitopes that had not been reported from the SARS-CoV-2-infected A549 cells and HEK293T cells, and at the same time, we found that a

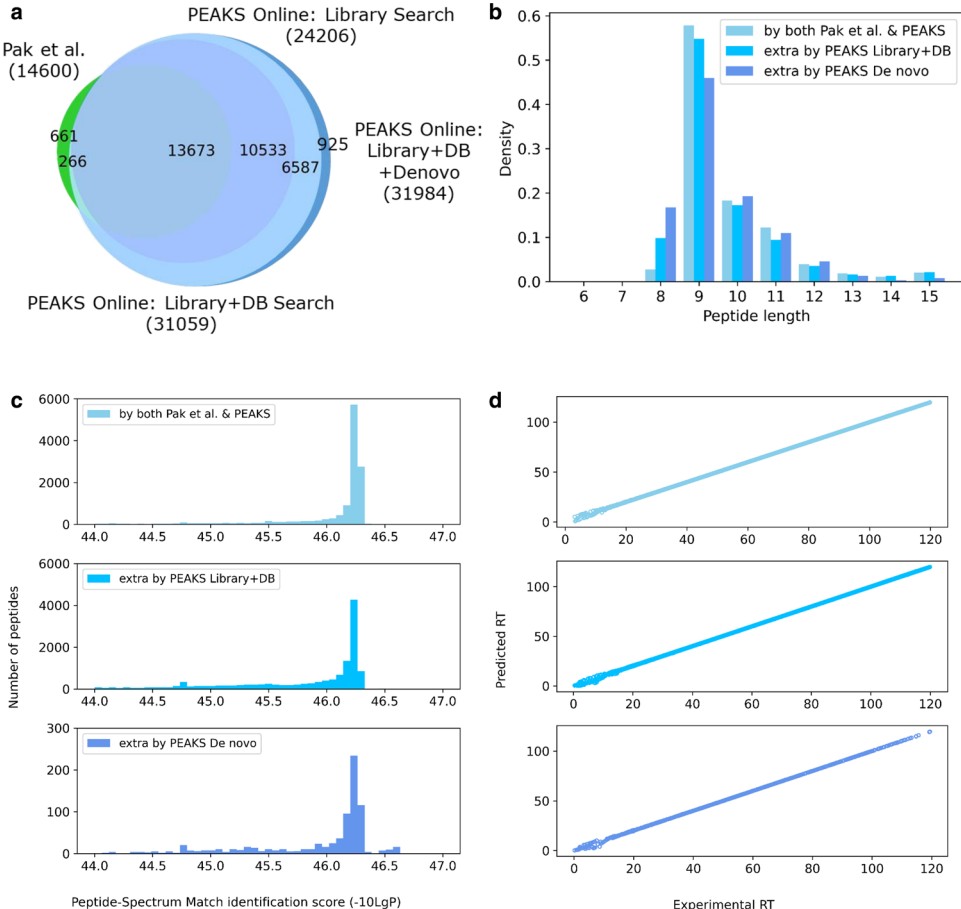

**Fig. 3 Integrated workflow for DIA analysis of the immunopeptidomics sample RA957. a** Venn diagram of the peptides identified by PEAKS Online spectral library search, database search, and de novo sequencing versus the results reported by Pak et al. [21]. **b** The length distributions of the identified peptides. **c** The distributions of the peptide-spectrum match (PSM) identification scores reported by PEAKS Online (named "-10LgP"). **d** The distributions of experimental retention times (RTs) versus predicted RTs of the identified peptides. The colors follow the legends in **c**.

previously reported T-cell epitope might be a false positive identification. Thus, both sensitivity and specificity offered by our platform are crucial in MS-based immunnopeptidomics for identifying the correct targets and reducing the number of uncertain candidates for vaccine development.

While the sensitivity improvement can be expected when multiple search engines or analysis methods are applied together, it is important to emphasize on their practicality and scalability, both of which are crucial for translational research and clinical applications. With PEAKS Online, one can easily perform complex analysis workflows in one single platform, without having to worry about which tools need to be used for what tasks, whether their input/output data are compatible, or how their results should be combined and interpreted. More importantly, the high-performance computing technology in PEAKS Online allows the high-throughput analysis of tera-scale datasets of thousands of samples for clinical applications on modern cloud computing platforms. It is very difficult, if not possible, to do such an analysis in a complex workflow that involves multiple tools from different sources. Thus, PEAKS Online offers high sensitivity, simplicity, and scalability. We are pretty confident that our platform will make a significant contribution to drive the proteomics research forward.

PEAKS Online uses random decoy peptides and spectra to estimate FDR. One possible limitation of this random decoy approach is that random peptides are too random while many false-positive HLA peptides have the correct anchor residues.

Perhaps a more constrained randomization can be applied so that anchor residues are fixed and other residues are permuted. However, a comprehensive evaluation on several benchmark datasets is needed in order to assess which decoy approach among randomization, constrained randomization, or two-species library[13,35], is more suitable for HLA peptides.

Currently PEAKS Online DIA supports three common variable modifications, including N-terminal (Acetylation), M(Oxidation), and NQ(Deamidation). It may be difficult to configure any number of post-translational modifications (PTMs) like in the case of DDA data. This is a common limitation for DIA analysis tools, because the libraries and the prediction models depend on the input data that were used to build them. It is difficult to generalize the models so that they are able to predict for new types of PTMs that they have not seen in the training data, as different types of PTMs tend to have different spectrum and RT distributions. On the other hand, PEAKS Online DDA can handle any number of fixed and variable PTMs. In fact, the computing power provided by PEAKS Online shall be tremendously beneficial for PTM search. We hope to provide a comprehensive assessment of PEAKS Online performance on PTMs in a future study.

## Methods

**De novo sequencing**. Our deep learning model DeepNovo for de novo sequencing of both DDA and DIA data was first proposed in refs. [23,26,27]. First, precursor features are detected together with their m/z, charge, retention time, and intensity profile from the LC-MS map. Next, for each precursor, we collect all MS/MS

**a**

| Peptide | Protein | Confidence score | MHC ligand assays Positive / All | HLA alleles | T cell assays Positive / All | IEDB link |
|---|---|---|---|---|---|---|
| QLTPTWRVY | S | 23.93 | 1 / 1 | C*16:01 | 1 / 1 | http://www.iedb.org/epitope/1323406 |
| NAPRITFG | N | 27.25 | | | | |
| APRITFGGP | N | 19.09 | 1 / 1 | B*07:02 | | http://www.iedb.org/epitope/1330992 |
| SIIAYTMSL | S | 20.28 | 6 / 6 | A*02:01, C*01:02 | 7 / 13 | http://www.iedb.org/epitope/1309137 |
| HLVDFQVTI | ORF6 | 17.57 | 10 / 10 | A*02:01, A*02:02, A*02:03, A*02:06, A*68:02 | 7 / 13 | http://www.iedb.org/epitope/24313 |
| NLIDSYFVV | nsp12 | 18.27 | 1 / 1 | A*02:01 | 1 / 1 | http://www.iedb.org/epitope/1313213 |

**b**

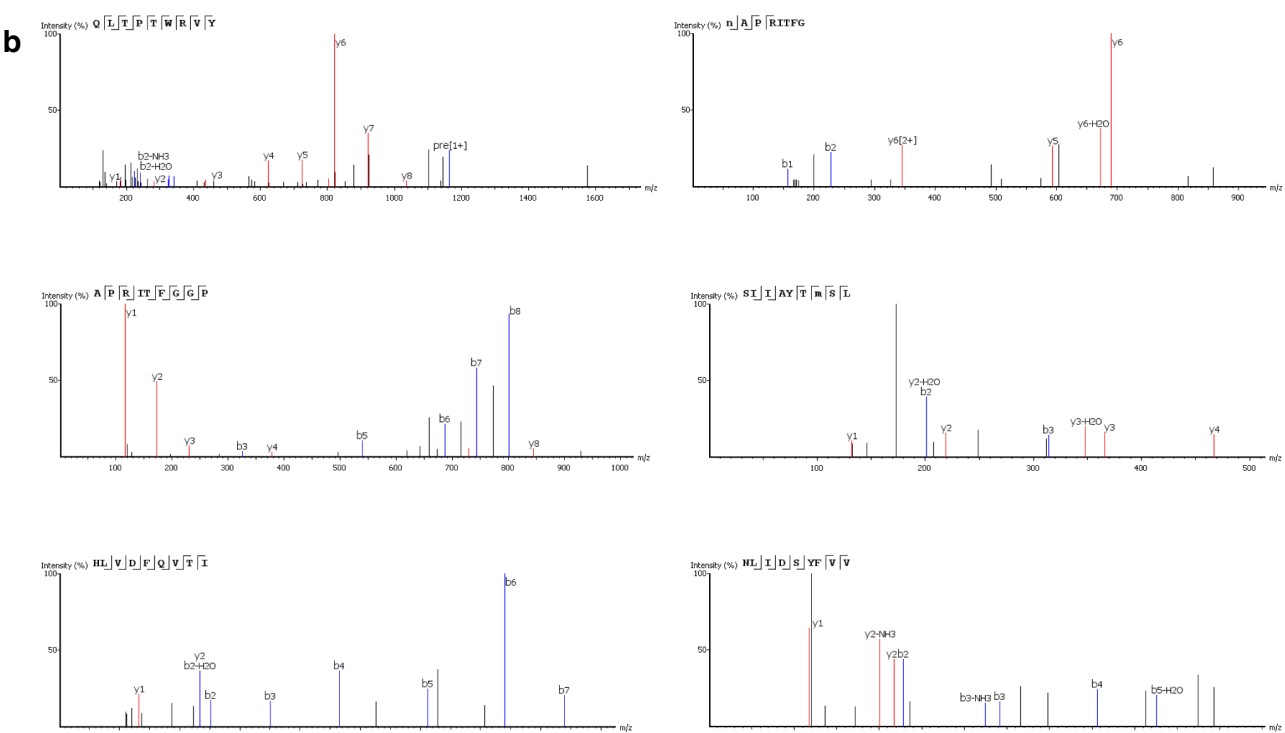

**Fig. 4 New T-cell epitopes identified by our platform from SARS-CoV-2-infected A549 cells and HEK293T cells. a** Detailed information of the T cell epitopes. **b** Peptide-spectrum matches of the T cell epitopes. (MHC: Major Histocompatibility Complex; HLA: Human Leukocyte Antigen; IEDB: Immune Epitope Database).

spectra so that they are within the precursor's retention-time and m/z ranges. For DIA data, there may be more than one spectrum and the number of spectra collected for a precursor may vary, so we select a fixed number of spectra that are closest to the center of the precursor's retention time. Then, we feed the precursor and its associated MS/MS spectra into DeepNovo neural networks to learn (i) the 3D shapes of fragment ions along m/z and retention time dimensions, (ii) the correlation between the precursor and its fragment ions, and (iii) the peptide sequence patterns. Our de novo sequencing framework predicts a peptide sequence by iteratively predicting one amino acid after another. At each iteration, two classification models are combined to predict the next amino acid by conditioning on the output of previous steps. The first model uses convolutional neural networks to learn patterns of the precursor and fragment ions. The second model uses long-short term memory recurrent neural networks to learn patterns of the peptide sequence. Finally, de novo peptides can be validated through an augmented database search with a controlled false discovery rate (FDR) to ensure that they are supported by significant peptide-spectrum matches.

**DDA database search**. PEAKS DB database search algorithm uses the state-of-art "denovo assisted" strategy to achieve both sensitivity and specificity[9]. Taking advantage of existing de novo sequencing results, PEAKS DB first performs a tag search against the protein sequences using the de novo sequencing results from the previous step. The tag search starts with the seeds constructed from de novo sequences then tries to extend the seed to the whole peptide sequence matched with the protein sequences. This method is fast enough allowing us to filter a full fasta sequence database such as NBCI (billions of proteins) to a set of proteins which is only related to the data set in a reasonable amount of time. Then a small set of high

quality spectra are searched against the filtered protein set. The match results are used to calibrate the mass error and train the scoring model. In the last step, all the spectra are searched against the filtered protein set. The pre-trained LDA model is used to separate the target and decoy matches. The overlap of the de novo sequencing result and peptide from the protein database for the same spectrum gives a good indication whether the peptide is a true match or not. By adding this as a feature in the LDA model, it significantly boosts the performance of the search engine.

**DIA spectral library search**. Initially, we tried the two-species library method introduced by DIA-NN[35] for FDR estimation so that fair comparisons can be made among different softwares. However, we found that using spectral libraries built from other species as the decoy might violate the underlying assumptions of target-decoy search strategy and lead to biased results. This is because peptides identified from different species tend to have different retention time distribution and amino acid frequencies. Modern DIA search engines like DIA-NN include the above features in its on-the-fly-training procedure, which means, information regarding whether a peptide is decoy is leaked to the classifier. To alleviate this problem, we propose to generate random decoys. For each spectrum in the target library, a decoy spectrum is generated by randomly permuting the peptide sequence and its corresponding fragment ions. The retention time (RT) is randomly sampled from the target peptides' RT distribution.

PEAKS DIA first performs feature detection on raw DIA data to find peptide features and associate their corresponding MS/MS spectra. In the meantime, it also creates a fake peptide feature for each peptide in the spectral library to enhance sensitivity. Then for each peptide feature, we can filter the spectral library by

precursor m/z and charge to find the potential library spectra that match the feature. Next, PEAKS DIA conducts two rounds of spectral library search. In the first round search, a rank normalized inner product score is given to each potential library spectrum and peptide feature pair. The target peptide features identified at 0.1% FDR are then selected to build an RT alignment model between the observed RT and indexed RT (iRT) stored in the library. The best precursor error tolerance and fragment ion error tolerance are obtained by a grid search on a small subset of peptide-feature-library-spectrum. In the second round spectral library search, PEAKS DIA can further reduce the search space of library spectra by filtering along the RT dimension. The match between a library spectrum and a peptide feature's associating MS/MS spectra is evaluated again in a sliding window fashion, with a window size of 7. The window slides over the RT dimension, each window containing seven consecutive MS2 spectra. We use weighted rank normalized inner product to evaluate the similarity between a window of MS/MS spectra and a library spectrum. The library spectrum with the best matched window is retained for each peptide feature. PEAKS DIA then computes 37 hand crafted features for each peptide-feature-library-spectrum match (Supplementary Table 1). We pass these features to a feedforward neural network classifier to perform on the fly training so that PEAKS DIA can better distinguish between target and decoy matches.

**DIA direct database search**. For the features that are not identified in the spectral library search step, PEAKS DIA then performs database search against protein sequences from a user provided fasta database. A protein sequence is first cleaved into peptides according to the enzyme/cleavage rules specified by users. Next, the peptide search space is filtered by peptide features' m/z and charge, so that only the peptides that can potentially match to a peptide feature are retained. PEAKS DIA then applies an internal spectrum prediction model and iRT prediction model on these peptides and builds a predicted spectral library. Finally, PEAKS DIA conducts spectral library search on the generated spectral library and the FDR is controlled using a non-parametric q-value estimation method.

**Spectrum and iRT prediction by deep learning**. PEAKS DIA adopts a 4-layer transformer model to predict the theoretical spectrum. Each transformer layer has 256 hidden units. At each possible fragment position, the model will predict the theoretical intensities for the following eight fragment ions: b, b-H2O, b-NH3, b2+, y, y-H2O, y-NH3, y2+. The iRT prediction model in PEAKS DIA consists of three transformer layers followed by three residue convolution layers and a global maxpooling layer. The spectrum and iRT prediction models are briefly depicted in Supplementary Fig. 2.

**Refine spectral library by deep transfer learning**. The performance of library-based DIA analysis relies largely on the quality of the spectral library being used. Because being often acquired from different MS instrument settings, the library spectrum from a public library and the query DIA spectrum display different fragment ion intensity patterns and iRT values. To alleviate the performance deterioration using a public spectral library, we use deep transfer learning to adaptively refine the public library. Specifically, our method consists of two rounds of library search coupled with deep transfer learning for spectrum prediction. It starts with training a deep learning model for fragment ion intensity and iRT prediction using the provided public spectral library. The first-round search is then conducted to search DIA data against the public library. The resulting high-quality identifications are used to fine-tune the previously trained deep learning model via transfer learning. Next, the fine-tuned model is used to predict/refine fragment ion intensities and iRTs for the precursors in the initial public library. Finally, the second-round library search is performed with the refined public library, which is expected to have adapted to the MS instrument setting in which the query DIA data was measured.

**Rescoring database search results with deep learning-predicted spectra**. PEAKS Online originally uses 18 hand-selected features to evaluate the quality of a peptide-spectrum match (PSM). A pre-trained linear discriminant function is applied on the 18 features to generate a score for each PSM. The best PSM for each spectrum is selected and the PSMs are then sorted and filtered with FDR estimation. Alternatively, we can also export all candidate PSMs for each spectrum and then use Percolator to rescore the PSMs and improve the identification. Previous studies have shown that, for HLA peptides, adding additional features generated from deep learning-predicted spectra can significantly increase the number of identified HLA peptides[22,37]. Here we include three spectrum similarity features, namely Pearson correlation, cosine similarity, and spectra angle between the predicted and the observed spectra to the Percolator rescoring step.

**The distributed high-performance computing architecture of PEAKS Online**. Our PEAKS Online platform is built with the Akka Toolkit, which is a general purpose framework to create reactive, distributed, parallel and resilient software systems. Akka implements the Actor Model on the JVM, and is widely used in the industry for cloud computing applications. We built our own distributed computing architecture on top of it utilizing its message-based inter-actor communication and reactive data streaming capabilities. We designed a single-master

multi-workers computing cluster where the master node takes care of maintaining system state and scheduling tasks, while workers get the computation done. All these nodes are Akka actors and the worker nodes communicate with the master node to synchronize their states, receive tasks, and submit results by exchanging messages.

To achieve high parallelism and best performance, we divide each analytical project into computation units by data fractions and multiple fractions can run concurrently on different worker nodes. Inside each fraction, we also take advantage of Akka's reactive data streaming to utilize CPU multi-threading to speed up computation further. At the end, distributed datasets are merged together to produce final results presented to users.

Because the amount of data we are dealing with is huge, as well as the distributed nature of our system, we also integrated a distributed data storage, Apache Cassandra, in our platform to store both intermediate and final results. We picked Cassandra for its asynchronous masterless design and linear scalability. So there's no performance bottleneck on individual data nodes and low latency and high throughput can be achieved even with our enormous data size. Because all nodes are connected to Cassandra and all data is persisted into it, worker nodes can share and access data created by others. This allows us to hand over tasks among nodes along the computation process and perform data aggregation in the end.

**Reporting summary**. Further information on research design is available in the Nature Research Reporting Summary linked to this article.

## Data availability
The dataset from Muntel et al. [31] is available in the PRIDE database under accession code PXD013658. The dataset from Xuan et al. [32] is available in the MassIVE database under accession code MSV000084976. The dataset from Sarkizova et al. [36] is available in the MassIVE database under accession code MSV000084172. The dataset from Bassani-Sternberg et al. [14] is available in the PRIDE database under accession code PXD004894. The dataset from Pak et al. [21] is available in the PRIDE database under accession code PXD022950. The dataset from Weingarten-Gabbay et al. [19] is available in the MassIVE database under accession code MSV000087225.

## Code availability
PEAKS Online is available for academic users; the requests can be sent to the corresponding author, Dr. Ming Li. For public usage, PEAKS Online is provided as a web service which can be accessed with the following link: https://peaksonline.bioinfor.com. The source code is publicly available on GitHub[42] (https://github.com/lxinbsi/peaks-online) and Zenodo (https://doi.org/10.5281/zenodo.6529062).

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

## Acknowledgements

This work was funded in part by the NSERC OGP0046506 grant (for M.L.), the Canada Research Chair program (for M.L.), and the Innovative and Entrepreneur Team Program of Zhejiang, number 2019R02002 (for M.L). We thank Dr. Roman Fischer and Dr. Philip Charles from the University of Oxford for providing data for large-scale testing. We thank Dr. Tharan Srikumar from Bruker Daltonics for providing timTOF testing data. We thank Dr. Sven Brehmer from Bruker Daltonics for providing support for Bruker software dependency. We thank Dr. Lei Guo for the advice on the initial architecture. We thank Ms. Wenting Li from Dougu Information Technology, China for application testing.

## Author contributions

L.X. and B.S. contributed to software function design; L.X. and S.P. contributed to software architecture design; L.X., R.Q., X.C., N.H.T., and H.B. contributed to algorithm design and implementation; S.R. and X.H. contributed to software implementation; B.M. contributed to software testing; B.S. contributed to result validation; L.X., R.Q, X.C., N.H.T., B.S., and M.L. wrote and revised the manuscript; B.S. and M.L. supervised the study.

## Competing interests

L.X., R.Q., X.C., H.T., S.P., S.R., H.B., X.H., B.M., and B.S. are employees of Bioinformatics Solutions Inc. The other authors declare no competing interests.
