## [Peer Review File · Nature Communications]

REVIEWER COMMENTS

Reviewer #1 (Remarks to the Author):

The manuscript presents a streamlined platform PEAKS Online for both DDA and DIA proteomics data by integrating spectral library search, database search, and de novo sequencing. The authors compared PEAKS Online (POL) with DIA-NN, Spectronaut and MaxQuant on the benchmark datasets, and reported the better sensitivity of POL.

Overall, I think it is a valuable one-stop platform to improve the sensitivity of peptides identification using different approaches. However, I think the authors should address the major questions and comments detailed below.

1. 5~30% more peptides were identified by POL. However, the results themselves are not unexpected. It was discovered that more peptides will be identified when multiple search engines or analysis methods are applied together. As the three analysis approaches used in PEAKS Online have been well published previously, the significance or the novelty of this manuscript is unclear.

2. In the workflow of PEAKS Online, the FDR control was estimated in each step. The risk of false-positive is high when the searching space is huge. Besides the overall FDR control, the authors should also check the reliabilities or confidence of PSMs obtained from spectral library search, database search, and de novo sequencing separately. The results given by the different approaches are comparable? For example, they may take a look at the distributions of the PEAKS Online scores of PSMs reported by different approaches. Should we use a more stringent FDR control in de novo sequencing?

3. The figures summarizing the results should be adjusted to make them more visible.

Reviewer #2 (Remarks to the Author):

The paper describes a sophisticated cloud based pipeline dubbed PEAKS-online for both DDA and DIA MS data analysis based the commercial PEAKS software. The authors need to explain certain aspects more clearly and provide more details about the pipeline and the results.

Major:

- "The platform is also implemented with the latest distributed high-performance computing technology". Could you describe briefly in the main text and more detailed in the supplementary which technologies you use, how the parallelization is done?

- I could not find information in the text how a user would run the pipeline. Will it be provided as a webservice, a docker image or a preinstalled service on AWS or other cloud computing providers? Do you support all major mass spec vendor formats? Are the workflows configurable, i.e. can the user define which step he/she wants to execute in which order? If yes how can this configuration be done? How will the results be reported?

- Is there any way the reader can test PEAKS online?

- FDR. The calculation and control of the global FDR in complex workflows is not trivial. In the text you mention that "while still controlling the overall FDR". How do you control the global or overall FDR of the final list of identified peptides? Are the local FDRs of in each step adapted if you set the global FDR and if yes how?

- In Figure 1c & d you estimate the FDR using 2 species library approach, as indicated in the x-axis label. However in the supplementary you state that this approach is not feasible since the DL model is able to detect which peptides are from which species and you use random peptides instead. Could you please explain which decoy approach you used for the DIA library search in Fig1c/d?

- In Figure 1f, if i understand right, you used PEAKS DB for database peptide search in PEAKS online, built a library and searched this library, and then performed de novo as indicated in the supplementary "Application in cancer immunopeptidomics"? However in the main text "The standard database search results ..." seems to indicate that you only performed a database search. Could you better describe how you performed this comparison?

- In Figure 1 f could you break down the Mel15 results into different alleles, peptide lengths and charge states in a suppl. figure?

- Could you also compare PEAKS online to the most recent version of MS2Rescore for DDA (Mel-15, A0204 and A0206) in Figure 1f?

- In the Abstract and Introduction you state that for immunopeptidomics samples the increase was 2-fold. This is only correct if you do not use rescoring for MQ and this should be clearly mentioned in the text.

- Random spectra and peptides also have their disadvantages since they can be too random and lack correlations present in real spectra or peptides. This could be a problem especially for HLA peptides, which have specific anchor residues. It is not clear to me why random peptides could not encounter a similar problem as peptides from a different species, when you perform the on the fly training in PEAKS DIA (the RT may not be an issue, but other feature scores may be biased). Many false positives in HLA peptides have the correct anchor residues, but the other amino acids of the peptide may be wrong, and these false positives are not well reflected by random sequences. How do you test whether your FDR is actually more or less correct? A simple way to go may be to show that the distribution of the neural net output scores of random peptides matches indeed the low score end of the NN score distribution of all target peptides.

- You say that PEAKS DIA predict spectra for a database search. Could you elaborate a bit more on the spectrum prediction model (a graphical depiction of the model may help)? How does this model compare to prosit or MS2PIP? Which PTMs can you include in the prediction, which charge states and which peptide lengths? Is the spectrum prediction trained on tryptic peptides and HLA-binding peptides separately, or do you use transfer learning for HLA spectra?

- You do not address whether and how PTMs can be included in your workflows. Can one just configure any number of fixed and variable PTMs in PEAKS DIA or do you rely on PEAKS denovo for PTM search?

- The length distribution in Suppl Fig S1 looks weird. 7-mers are highly unlikely to bind to HLA complexes, and there seems to be an elevated number of 8-mers, too. Also the rt correlation in Fig S1 doesn't look great compared to Pak et al. and the sequence motifs look worse compared to the database search. It is possible that you have higher error in your denovo search than estimated by the FDR? Could you elaborate which peptides you identify by denovo (charge state) and show some examples together with their spectra. How many of these denovo peptides can be confirmed by rescoring compared to peptides found by sequence DB search?

Minor:

- In the supplementary section you describe in detail which software are used in each step. It would be nice to have an overview figure in the supplementary that show which software are used in which steps and how the different steps can be connected (flow chart). It would also be nice to have a table with all software solutions used, their publications and github links (if open source) or other download links.

- "In the meantime, it also creates a fake peptide feature for each peptide in the spectral library to enhance sensitivity." I don't quite understand this sentence.

- ".. sliding window fashion, with a window size of 7 ..". The window slides over RT or mass? What is the unit of the window size (seconds)?

- "PEAKS Online originally uses 18 hand-selected features to evaluate the quality of a peptide-spectrum match (PSM)". Could you provide a list of these 18 features and also of the 28 features you use for spectrum matching?

- "How many nodes, CPUs and memory did you use on AWS for the results in Table 1". How does the actual computation time decrease with increasing number of nodes or CPUs?

- Supplementary : "Previous studies have shown...": please provide references.

Reviewer #3 (Remarks to the Author):

First, the reviewer would like to answer some questions:

What are the noteworthy results?

A unified framework achieved a two-fold increase in HLA peptide identification than previously reported results.

High-performance computing performed high-throughput analysis on terabytes of data.

Will the work be of significance to the field and related fields? How does it compare to the established literature? If the work is not original, please provide relevant references.

Yes, the work is of significance. Other tools have not built up such a unified framework processing DDA and DIA through spectral library search, database search, and de novo sequencing.

Does the work support the conclusions and claims, or is additional evidence needed?

Yes, it does.

Are there any flaws in the data analysis, interpretation and conclusions? Do these prohibit publication or require revision?

Yes, please see the below major and minor suggestions. If the authors can address these issues, it is ready for publication.

Is the methodology sound? Does the work meet the expected standards in your field?

Yes, it sounds good, and meet the expected standards.

Is there enough detail provided in the methods for the work to be reproduced?

Yes, it is.

The authors demonstrated that PEAKS Online is an ultra-sensitive and streamlined platform for DDA and DIA mass spectrometry analysis. It integrates deep learning-based solutions of

spectral library search, database search, and de novo sequencing under a unified framework,

which boosts the sensitivity and specificity of peptide identification. It shows that deep learning can unify the analysis pipelines and improve the performance of peptide identification, and HPC can process terabytes of data efficiently. This beautiful work shows the power of Bioinformatics.

However, the reviewer does have some major and minor suggestions.

Here are the major suggestions.

1. Since PEAKS Online is a HPC version, it is better to show the performance of HPC (i.e. for the same terabytes of data, draw a figure of processing time vs number of CPU cores).
2. Since database search and spectral library search of PEAKS Online both use target-decoy approach for FDR control, it is better to do the null search to prove that there is no bias in the design of decoy method. In the null search, each precursor mass can be shifted by adding a big mass (e.g. 100 Da). The shifted MS/MS file (e.g. MGF format) can be processed via database search and spectral library search. Draw a figure of score distributions of target and decoy PSMs for database search and spectral library search, respectively.
3. One difficulty to identify HLA peptidomes is that the peptides are short (many are in length of 8-12 AA) and fragment ions may be not enough to discriminate similar sequences by conventional database search. In this case, deep learning can help because it can incorporate more features (i.e. rescoring). However, the results of deep learning still need further validation (e.g. using the synthetic HLA peptides). The authors may show one example that one PSM identified by Pak, et al. (2021) and by PEAKS Online have different sequences, and the sequence by PEAKS Online can be validated by synthetic peptides (i.e. they have similar MS/MS spectra).
4. PEAKS Online shows better performance than other tools on unmodified peptides/proteins. How about PEAKS Online on PTMs (e.g. phosphorylation data)? The community also expects deep learning can help improving identification and localization of PTMs.

Here are the minor suggestions.

1. Please explain 'two-species library method' in Figure 1c, 1d, and Figure 2a.
2. Why all peptides with 7 AA are new in Figure 3d? Is it because that the previous method excludes length 7?
3. Supplementary Figure S3a looks like a mixture spectrum.
4. In main text of 'Competing Interests Statement', it is 'The authors declare no competing interests'. It needs to be revised according to 'Editorial Policy Checklist'.

Point-by-point response letter

We thank the Editor and the Reviewers for your constructive comments that greatly helped us to improve our manuscript. We have fully addressed all concerns raised by the Editor and the Reviewers, including FDR calculation, additional validation of our immunopeptidomics analysis results, and all other technical concerns. The point-by-point responses to all questions of the Reviewers can be found below. All revision changes in the manuscript are marked in blue color for your convenience.

Reviewer #1

The manuscript presents a streamlined platform PEAKS Online for both DDA and DIA proteomics data by integrating spectral library search, database search, and de novo sequencing. The authors compared PEAKS Online (POL) with DIA-NN, Spectronaut and MaxQuant on the benchmark datasets, and reported the better sensitivity of POL. Overall, I think it is a valuable one-stop platform to improve the sensitivity of peptides identification using different approaches. However, I think the authors should address the major questions and comments detailed below.

Reviewer #1 - question #1:

5~30% more peptides were identified by POL. However, the results themselves are not unexpected. It was discovered that more peptides will be identified when multiple search engines or analysis methods are applied together. As the three analysis approaches used in PEAKS Online have been well published previously, the significance or the novelty of this manuscript is unclear.

Authors' response:

As Reviewer #1 has pointed out in the summary, PEAKS Online is a valuable one-stop platform to improve the sensitivity of peptide identification by integrating DDA, DIA, spectral library search, database search, and de novo sequencing. While the sensitivity improvement can be expected and some research ideas or methodologies have been published, we would like to emphasize on their practicality and scalability, both of which are crucial for translational research and clinical applications.

With PEAKS Online, one can easily perform complex analysis workflows in one single platform, without having to worry about which tools need to be used for what tasks, whether their input/output data are compatible, or how their results should be combined and interpreted. More importantly, the high-performance computing technology in PEAKS Online allows the high-throughput analysis of tera-scale datasets of thousands of samples for clinical applications on modern cloud computing platforms. It is very difficult to do such a large-scale analysis when using tools from different sources. Thus, PEAKS Online offers high sensitivity, simplicity, and scalability. We are pretty confident that our platform will make a significant contribution to drive the proteomics research forward. We have added these arguments in the Discussion in the main text, line 271.

Reviewer #1 - question #2:

In the workflow of PEAKS Online, the FDR control was estimated in each step. The risk of false-positive is high when the searching space is huge. Besides the overall FDR control, the authors should also check the reliabilities or confidence of PSMs obtained from spectral library search, database search, and de novo sequencing separately. The results given by the different approaches are comparable? For example, they may take a look at the distributions of the PEAKS Online scores of PSMs reported by different approaches. Should we use a more stringent FDR control in de novo sequencing?

Authors' response:

We agree with Reviewer #1 that FDR control is the most important problem when the searching space is huge and different analysis approaches are integrated. In this revised manuscript, as depicted in Figure 1b, we collected the list of peptides identified by the three approaches (spectral library search, database search, and de novo sequencing), built a spectral library from this peptide list, and performed a final search of the whole dataset against this new library. This

final search was meant to re-confirm the identified peptides and to provide a unified global FDR of 1%. We found that the number of identified peptides from the DIA dataset RA957 were reduced by about 11% than previously reported in our initial manuscript (Figure 3a).

Supplementary Figure S6 shows the PSM score distributions of random decoy peptides and the target peptides. As expected for FDR control, the score distribution of the random decoy peptides was indeed located at the lower end of the score distribution of the target peptides.

We also followed Reviewer #1's advice to check whether the identification results reported by different approaches were comparable. Figure 3 shows the breakdown details of the peptides identified by both Pak et al. and PEAKS Online, and the new peptides identified by PEAKS Online spectral library search, database search, and de novo sequencing separately. We found that the distributions of peptide lengths, PSM scores, and retention times (RTs) of the peptides reported by different approaches were comparable. In addition, the predicted RTs were highly correlated with the experimental RTs. Since this DIA dataset RA957 is HLA data, we also ran NetMHCpan on all identified peptides by PEAKS Online and found that about 88% of them were predicted as weak binders (rank \leq 2%), 76% as strong binders (rank \leq 0.5%). The complete details of all PSMs identified by PEAKS Online are provided in Supplementary Table S4.

We also followed Reviewer #1 advice to apply more stringent control on de novo sequencing results. First, we only selected peptides with positional confidence scores of at least 90 (in a scale of 0-100), which means that every amino acid was supported by both b- and y- ions in our de novo sequencing algorithm. Second, the selected de novo peptides were further subjected to the final search with a global FDR of 1% as mentioned above. As shown in Figure 3, the de novo peptides had similar distributions of peptide lengths, PSM scores, and RTs as those identified by spectral library search or database search. There was a slightly elevated number of 8-mers in the length distribution of the de novo peptides. Supplementary Figure S7 shows the PSMs of some example de novo peptides together with their precursor profiles and fragment ion profiles. To validate those de novo peptides, we also ran an independent spectrum prediction tool MS²PIP and found that the predicted spectra were highly correlated with the experimental spectra (Supplementary Figure S7d). The amino acids at the anchor positions were consistent with the binding motifs of the sample HLA alleles (Supplementary Figure S7e).

Reviewer #1 - question #3:

The figures summarizing the results should be adjusted to make them more visible.

Authors' response:

We have revised all figures to increase their sizes and text labels. We have also updated their resolutions to 600 dpi according to the journal guidelines. Please let us know if you still find any figures not clear enough and we shall adjust them accordingly.

Reviewer #2

The paper describes a sophisticated cloud based pipeline dubbed PEAKS-online for both DDA and DIA MS data analysis based the commercial PEAKS software. The authors need to explain certain aspects more clearly and provide more details about the pipeline and the results.

Reviewer #2 - question #1:

"The platform is also implemented with the latest distributed high-performance computing technology". Could you describe briefly in the main text and more detailed in the supplementary which technologies you use, how the parallelization is done?

Authors' response:

We have followed Reviewer #2 advice to provide a brief description of our high-performance computing platform in the main text, line 127 and a summary diagram in Supplementary Figure S3. More details of the technologies and the parallelization are provided in the Supplementary Information, line 124. We copied the brief description below for your convenience.

PEAKS Online is designed to achieve high parallelism, high scalability, and high stability with the modern distributed high-performance computing technology (Supplementary Figure S3). The platform is built on top of the Java Akka Toolkit. We implemented a master-workers computation cluster with Akka's actor based messaging model. Following the map-reduce pattern, we split larger projects into fractions, run them in parallel on different worker nodes, and aggregate the results in the end to present to the user. A distributed data storage, Apache Cassandra, is integrated in our system for data persistence. So datasets produced by different worker nodes can be shared and accessed, which allows us to hand over tasks between workers and perform data aggregation in the end. More details of PEAKS Online's architecture can be found in the Supplementary Information.

Reviewer #2 - question #2:

I could not find information in the text how a user would run the pipeline. Will it be provided as a webservice, a docker image or a preinstalled service on AWS or other cloud computing providers? Do you support all major mass spec vendor formats? Are the workflows configurable, i.e. can the user define which step he/she wants to execute in which order? If yes how can this configuration be done? How will the results be reported?

Authors' response:

A full documentation (i.e. user manual) of PEAKS Online is included in the Supplementary Information.

PEAKS Online is a vendor neutral computing platform and supports all major mass spectrometry data formats, including mzXML, mzML, Thermo .raw files, Waters raw folders, Bruker .d folders, Thermo DIA, Bruker DIA, WIFF DDA/DIA.

PEAKS Online is designed to be cross platform. It can be deployed on any OS which supports Java with a pre-compiled installation package. For public usage, it will be provided as a web service which can be accessed with the following link: <https://peaksonline.bioinfor.com/>. Each user will need an account to access all the functions which will be assigned upon request.

The workflows are configurable. Each step can be enabled/diabled easily through a web page. The results can be accessed through the result web pages. They also can be downloaded as csv files or other formats if needed.

For more information, please refer to the full documentation in the Supplementary Information.

Reviewer #2 - question #3:

Is there any way the reader can test PEAKS Online?

Authors' response:

We have provided 4 accounts in the cover letter to the Editor so that the Editor and the Reviewers can test PEAKS Online. PEAKS Online is also available for academic users; the requests can be sent to the corresponding author, Dr. Ming Li. For public usage, PEAKS Online is provided as a web service which can be accessed with the following link:

<https://peaksonline.bioinfor.com/>. We have added this information of software availability in the main text, line 307.

Reviewer #2 - question #4:

FDR. The calculation and control of the global FDR in complex workflows is not trivial. In the text you mention that "while still controlling the overall FDR". How do you control the global or overall FDR of the final list of identified peptides? Are the local FDRs of in each step adapted if you set the global FDR and if yes how?

Authors' response:

As all three Reviewers have pointed out, FDR control is the most important problem in complex workflows where different analysis approaches are integrated. In this revised manuscript, as depicted in Figure 1b, we collected the list of peptides identified by the three approaches (spectral library search, database search, and de novo sequencing), built a spectral library from this peptide list, and performed a final search of the whole dataset against this new library. This final search was meant to re-confirm the identified peptides and to provide a unified global FDR of 1%. We found that the number of identified peptides from the DIA dataset RA957 were reduced by about 11% than previously reported in our initial manuscript (Figure 3a).

Supplementary Figure S6 shows the PSM score distributions of random decoy peptides and the target peptides. As expected for FDR control, the score distribution of the random decoy peptides was indeed located at the lower end of the score distribution of the target peptides.

The global FDR is recommended in complex workflows where different analysis approaches are integrated, such as the workflow in Figure 1b. The final peptides were those that passed the 1% global FDR. If any peptides passed the 1% local FDRs but failed the global one, they were not included in the final results.

Reviewer #2 - question #5:

In Figure 1c & d you estimate the FDR using 2 species library approach, as indicated in the x-axis label. However in the supplementary you state that this approach is not feasible since the DL model is able to detect which peptides are from which species and you use random peptides

instead. Could you please explain which decoy approach you used for the DIA library search in Fig1c/d?

Authors' response:

We apologize for this confusion. We used the random decoy approach to estimate FDR throughout our PEAKS Online platform, including the spectral library search and database search. In our initial experiments, when comparing to DIA-NN by Demichev et al. [35], we tried to follow their two-species library approach for a fair comparison. However, in the final experiments, we decided to use the random decoy approach because (i) as we discussed in the supplementary, deep learning models may be able to detect which peptides are from which species, and (ii) we need to keep the consistency of our entire platform where the random decoy approach is used for DDA and DIA, spectral library search and database search.

The evaluation results in Figures 1c/d were calculated using the random decoy approach. For DIA-NN and Spectronaut, we generated random decoy libraries, converted them into Spectronaut format, and supplied them to DIA-NN and Spectronaut. The mistake in the x-axis label was because we forgot to update the plotting scripts accordingly.

Reviewer #2 - question #6:

In Figure 1f, if I understand right, you used PEAKS DB for database peptide search in PEAKS online, built a library and searched this library, and then performed de novo as indicated in the supplementary "Application in cancer immunopeptidomics"? However in the main text "The standard database search results ..." seems to indicate that you only performed a database search. Could you better describe how you performed this comparison?

Authors' response:

We apologize for the confusion, we have revised the main text, line 144, and the Supplementary Information, line 147, to better describe the experiments. Basically there were two different analyses in the section "Applications in cancer immunopeptidomics". The first one was performed on three DDA immunopeptidomics datasets from Sarkizova et al. [36] and Bassani-Sternberg et al. [14]. The second one was performed on the immunopeptidomics sample RA957 from Pak et al. [21], which included both DDA and DIA data.

Figure 1f shows the performance evaluation of PEAKS Online versus MaxQuant-Prosit on three DDA immunopeptidomics datasets from Sarkozova et al. [36] and Bassani-Sternberg et al. [14]. This experiment was requested by the Editor during the initial assessment of our manuscript, where we needed to compare to an immunopeptidomics-optimized version of Prosit by Wilhelm et al. [22]. The three DDA immunopeptidomics datasets were selected because they were also used in Wilhelm et al. [22]. They are DDA data, so a standard database search by PEAKS Online or MaxQuant can be performed. However, since most search engines were not originally designed for HLA peptides, Wilhelm et al. [22] proposed to train Prosit, a deep learning model for spectrum prediction, on synthetic HLA peptides, and then used that model to rescore database search results to improve the peptide identification. We show in Figure 1f that, for both standard database search results and rescoring results, PEAKS Online performed better than MaxQuant and MaxQuant-Prosit, respectively. More details of the database search and rescoring procedure are provided in the Supplementary Information, line 152.

The second analysis was performed on the immunopeptidomics dataset RA957 from Pak et al. [21]. This dataset includes both DDA and DIA data. A DIA integrated workflow using spectral library search, library search, and de novo sequencing were performed with PEAKS Online, and the results were compared to those reported with MaxQuant-Spectronaut in Pak et al. [21]. The analysis results were presented in Figures 2 and 3 and were discussed in the main text, line 170. More details of the workflow and the analysis were provided in the Supplementary Information, line 169.

Reviewer #2 - question #7:

Could you also compare PEAKS online to the most recent version of MS2Rescore for DDA (Mel-15, A0204 and A0206) in Figure 1f?

Authors' response:

We have followed Reviewer #2's advice to compare PEAKS Online rescoring results to MS2Rescore for three DDA datasets Mel-15, A0204, and A0206 in Figure 1f. We found that they were comparable for two datasets Mel-15 and A0204, whereas for dataset A0206 MS2Rescore identified 17% more peptides. We have updated these comparison results in Figure 1f and in the main text, line 159.

Reviewer #2 - question #8:

In the Abstract and Introduction you state that for immunopeptidomics samples the increase was 2-fold. This is only correct if you do not use rescoring for MQ and this should be clearly mentioned in the text.

Authors' response:

Thank you for pointing out, this is an overstatement and not correct. We have revised the statements in the Abstract and Introduction as following:

“When evaluated on three DDA immunopeptidomics datasets, our standard database search identified 1.7-4.1 times more peptides, while our rescoring with deep learning-predicted spectra identified 1.0-1.4 times more peptides than previously reported results. In another evaluation on a DIA immunopeptidomics dataset, our integrated workflow of spectral library search, database search, and de novo sequencing together identified 1.4-2.2 times more peptides than previously reported results.”

We think the revised statements above accurately summarized the results for DDA and DIA immunopeptidomics data using different analysis approaches. More details can be found in the corresponding sections in the main text. Please let us know if you still think the statement needs to be revised further.

Reviewer #2 - question #9:

Random spectra and peptides also have their disadvantages since they can be too random and lack correlations present in real spectra or peptides. This could be a problem especially for HLA peptides, which have specific anchor residues. It is not clear to me why random peptides could not encounter a similar problem as peptides from a different species, when you perform the on the fly training in PEAKS DIA (the RT may not be an issue, but other feature scores may be biased). Many false positives in HLA peptides have the correct anchor residues, but the other amino acids of the peptide may be wrong, and these false positives are not well reflected by random sequences. How do you test whether your FDR is actually more or less correct? A

simple way to go may be to show that the distribution of the neural net output scores of random peptides matches indeed the low score end of the NN score distribution of all target peptides.

Authors' response:

To address the concerns regarding the score distributions of target and decoy PSMs, we have added Supplementary Figure S6 to test our FDR as the Reviewers suggested. Supplementary Figure S6 shows the PSM score distributions of random decoy peptides and the target peptides on the DIA dataset RA957 (Figure 3a). As expected for FDR control, the score distribution of the random decoy peptides was indeed located at the lower end of the score distribution of the target peptides.

Regarding the concern that random peptides are too random while many false-positive HLA peptides have the correct anchor residues, perhaps a more constrained randomization can be applied so that anchor residues are fixed and other residues are permuted. However, a comprehensive evaluation on several benchmark datasets is needed in order to assess which decoy approach among two-species library, randomization, or constrained randomization, is more suitable for HLA peptides. We have added these limitations in the Discussion in the main text, line 283.

Reviewer #2 - question #10:

You say that PEAKS DIA predicts spectra for a database search. Could you elaborate a bit more on the spectrum prediction model (a graphical depiction of the model may help)? How does this model compare to prosit or MS2PIP? Which PTMs can you include in the prediction, which charge states and which peptide lengths? Is the spectrum prediction trained on tryptic peptides and HLA-binding peptides separately, or do you use transfer learning for HLA spectra?

Authors' response:

We have added Supplementary Figure S2 to describe our deep learning models for spectrum and iRT prediction. We did not directly evaluate the predicted spectra to those produced by Prosit or MS2PIP. Instead, we compared the peptide identifications after rescoring the standard database search results by using the predicted spectra. The comparison of our rescoring results to Prosit and MS2Rescore were shown in Figure 1f. We also used MS2PIP web-server as an

independent spectrum prediction tool to validate some of our identified peptides, e.g. in Supplementary Figures S5 and S7.

Our prediction models can include three common variable modifications, including N-terminal (Acetylation), M(Oxidation), and NQ(Deamidation). In this manuscript, we started with some internal pre-trained models on tryptic peptides and re-trained them with HLA data from Sarkizova et al. [36]. For the evaluation in Figure 1f, the data of Mel-15, A*02:04, and A*02:06 were excluded from the training data.

Reviewer #2 - question #11:

You do not address whether and how PTMs can be included in your workflows. Can one just configure any number of fixed and variable PTMs in PEAKS DIA or do you rely on PEAKS denovo for PTM search?

Authors' response:

Currently PEAKS DIA supports three common variable modifications, including N-terminal (Acetylation), M(Oxidation), and NQ(Deamidation). We are continuously working to extend our libraries and our prediction models. However, we think it may be not possible to configure any number of PTMs like in the case of DDA data. This is a common limitation for DIA analysis tools, not just PEAKS DIA, because the libraries and the prediction models depend on the input data that were used to build them. We once had some discussion with the authors of Prosit, who also faced similar challenges about prediction models for PTMs. It is difficult to generalize the models so that they are able to predict for new types of PTMs that they haven't seen in the training data, because different types of PTMs tend to have different spectrum and RT distributions. We have added these limitations in the Discussion section in the main text, line 290.

Reviewer #2 - question #12:

The length distribution in Suppl Fig S1 looks weird. 7-mers are highly unlikely to bind to HLA complexes, and there seems to be an elevated number of 8-mers, too. Also the rt correlation in Fig S1 doesn't look great compared to Pak et al. and the sequence motifs look worse compared to the database search. It is possible that you have higher error in your denovo search than

estimated by the FDR? Could you elaborate which peptides you identify by de novo (charge state) and show some examples together with their spectra. How many of these de novo peptides can be confirmed by rescoring compared to peptides found by sequence DB search?

Authors' response:

Thank you for pointing out this mistake, yes, 7-mers should be excluded from the identification of HLA peptides.

Regarding your concern about the quality of de novo peptides, in this revised manuscript, we have applied a more stringent control on de novo sequencing results. First, we only selected peptides with positional confidence scores of at least 90 (in a scale of 0-100), which means that every amino acid was supported by both b- and y- ions in our de novo sequencing algorithm. Second, the selected de novo peptides were further subjected to the final search with a global FDR of 1% as mentioned above. As shown in Figure 3, the de novo peptides had similar distributions of peptide lengths, PSM scores, and RTs as those identified by spectral library search or database search. There was a slightly elevated number of 8-mers in the length distribution of the de novo peptides. Supplementary Figure S7 shows the PSMs of some example de novo peptides together with their precursor profiles and fragment ion profiles. To validate those de novo peptides, we also ran an independent spectrum prediction tool MS²PIP and found that the predicted spectra were highly correlated with the experimental spectra (Supplementary Figure S7d). The amino acids at the anchor positions were consistent with the binding motifs of the sample HLA alleles (Supplementary Figure S7e). The complete details of all PSMs identified by PEAKS Online, including the de novo ones, are provided in Supplementary Table S4.

Reviewer #2 - question #13.

In the supplementary section you describe in detail which software are used in each step. It would be nice to have an overview figure in the supplementary that show which software are used in which steps and how the different steps can be connected (flow chart). It would also be nice to have a table with all software solutions used, their publications and github links (if open source) or other download links.

Authors' response:

We have added the Supplementary Figure S1 to provide an overview of how different deep learning tools were used for each specific analysis task in our PEAKS Online platform. Publications and links are also included where available. For the spectrum and RT prediction models that we have developed internally, we don't have publications yet. Note that the figure only lists the tools that we use in our platform, certainly there are many other tools developed by other groups for the same tasks.

Reviewer #2 - question #14:

".. sliding window fashion, with a window size of 7 ..". The window slides over RT or mass? What is the unit of the window size (seconds)?

Authors' response:

The match between a library spectrum and a peptide feature's associating MS/MS spectra is evaluated again in a sliding window fashion, with a window size of 7. The window slides over the RT dimension, each window containing 7 consecutive MS2 spectra. We use weighted rank normalized inner product to evaluate the similarity between a window of MS/MS spectra and a library spectrum. The library spectrum with the best matched window is retained for each peptide feature.

Reviewer #2 - question #15:

"PEAKS Online originally uses 18 hand-selected features to evaluate the quality of a peptide-spectrum match (PSM)". Could you provide a list of these 18 features and also of the 28 features you use for spectrum matching?

Authors' response:

The list of 18 features for evaluating the quality of PSMs by PEAKS Database search is available in Zhang et al. [9]. The list of 37 features (updated from previous 28) for peptide-feature-library-spectrum matching by PEAKS DIA Spectral library search is provided in Supplementary Table S1.

Reviewer #2 - question #16:

"How many nodes, CPUs and memory did you use on AWS for the results in Table 1". How does the actual computation time decrease with increasing number of nodes or CPUs?

Authors' response:

For the results in Supplementary Table S2 (updated name), we used 512 CPU cores and approximately 1 TB of RAM. Since that experiment used AWS and it was quite costly, we were not able to repeat it multiple times to measure the scalability. We have added another experiment on a smaller dataset in the Supplementary Figure S4 to demonstrate the scalability of PEAKS Online. Supplementary Figure S4 shows how the computational time efficiently decreased with the increasing number of CPU cores on an example dataset of 56 samples, 672 MS runs, with a total size of near 1 TB. The analysis was finished in 17 hours when 512 CPU cores were used. We repeated the analysis using different numbers of CPU cores to test the scalability of PEAKS Online.

Reviewer #2 - question #17:

Supplementary : "Previous studies have shown...": please provide references.

Authors' response:

We have added two references by Wilhelm et al. [22] and Declercq et al. [37]. Both studies have suggested that, by training models on non-tryptic or HLA peptides, deep learning-predicted spectra and retention times can be used to rescore the PSMs and to improve the identification of HLA peptides for mass spectrometry-based immunopeptidomics.

Reviewer #3

The authors demonstrated that PEAKS Online is an ultra-sensitive and streamlined platform for DDA and DIA mass spectrometry analysis. It integrates deep learning-based solutions of spectral library search, database search, and de novo sequencing under a unified framework, which boosts the sensitivity and specificity of peptide identification. It shows that deep learning can unify the analysis pipelines and improve the performance of peptide identification, and HPC can process terabytes of data efficiently. This beautiful work shows the power of Bioinformatics. However, the reviewer does have some major and minor suggestions.

Reviewer #3 - question #1:

Since PEAKS Online is a HPC version, it is better to show the performance of HPC (i.e. for the same terabytes of data, draw a figure of processing time vs number of CPU cores).

Authors' response:

We have added Supplementary Figure S4 to demonstrate the scalability of PEAKS Online, how the computational time efficiently decreased with the increasing number of CPU cores on an example dataset of 56 samples, 672 MS runs, with a total size of near 1 TB. The analysis was finished in 17 hours when 512 CPU cores were used. We repeated the analysis using different numbers of CPU cores to test the scalability of PEAKS Online (Supplementary Figure S4). Another example analysis on a larger project of more than four thousands samples with 3.7 terabytes of data was run on the AWS cloud system (Supplementary Table S2). Since AWS was quite costly, we were not able to repeat it multiple times to measure the scalability for this example.

Reviewer #3 - question #2.

Since database search and spectral library search of PEAKS Online both use target-decoy approach for FDR control, it is better to do the null search to prove that there is no bias in the

design of decoy method. In the null search, each precursor mass can be shifted by adding a big mass (e.g. 100 Da). The shifted MS/MS file (e.g. MGF format) can be processed via database search and spectral library search. Draw a figure of score distributions of target and decoy PSMs for database search and spectral library search, respectively.

Authors' response:

To address the concerns regarding the score distributions of target and decoy PSMs, we have added Supplementary Figure S6 to test our FDR as the Reviewers suggested. Supplementary Figure S6 shows the PSM score distributions of random decoy peptides and the target peptides on the DIA dataset RA957 (Figure 3a). As expected for FDR control, the score distribution of the random decoy peptides was indeed located at the lower end of the score distribution of the target peptides.

Regarding the “null search”, honestly we are not sure how to do it. If we understand it correctly, Reviewer #3 suggested adding a big mass (e.g. 100 Da) to every fragment ion in the MS/MS spectra and using those shifted MS/MS spectra for database search and spectral library search. But then how to deal with the total peptide mass? If we also add 100 Da to the peptide mass, then the amino acid sequences of the random and target peptides need to be changed accordingly? If we keep the same peptide mass and the amino acid sequences, the masses of the theoretical fragment ions of both random and target peptides won't match those in the shifted MS/MS spectra.

We hope that our results in Supplementary Figure S6 above could address Reviewer #3's concern about FDR control for the target-decoy approach.

Reviewer #3, question #3.

One difficulty to identify HLA peptidomes is that the peptides are short (many are in length of 8-12 AA) and fragment ions may be not enough to discriminate similar sequences by conventional database search. In this case, deep learning can help because it can incorporate more features (i.e. rescoring). However, the results of deep learning still need further validation (e.g. using the synthetic HLA peptides). The authors may show one example that one PSM identified by Pak, et al. (2021) and by PEAKS Online have different sequences, and the

sequence by PEAKS Online can be validated by synthetic peptides (i.e. they have similar MS/MS spectra).

Authors' response:

We have followed the reviewer's suggestion to look into those PSMs where the peptides identified by Pak et al. and by PEAKS Online were different. We found 241 such PSMs and they were presented in the Supplementary Table S3. We further investigated one example PSM with MS/MS scan id 57611, fraction RA957_R03 in the Supplementary Figure S5. As our lab doesn't have the capability to synthesize peptides, we took an alternative approach to predict the spectrum by using an independent tool MS²PIP (which was suggested by another Reviewer with more than 90% correlation with real spectra [40]). The mirror plots between the experimental spectrum and the predicted spectra in the Supplementary Figure S5 show a Pearson correlation of 87% for the peptide identified by PEAKS Online, but only -4% for the one by Pak et al. We further used NetMHCpan to check the peptides' binding affinities. The peptide identified by PEAKS Online had a NetMHCpan rank of 0.0024, which was substantially stronger than that of the peptide identified by Pak et al. (1.4416, smaller is stronger). The second and last amino acids in the peptide identified by PEAKS Online clearly agreed with the binding motif for the allele. HLA-B*35:03. Overall, the Pearson correlation, the NetMHCpan rank and the binding motif indicated that the peptide identified by PEAKS Online was more accurate than the one by Pak et al.. We also found that, for 81% of 241 PSMs in the Supplementary Table S3, the peptides identified by PEAKS Online had stronger binding affinities than the ones by Pak et al., as predicted by NetMHCpan.

Reviewer #3 - question #4:

PEAKS Online shows better performance than other tools on unmodified peptides/proteins. How about PEAKS Online on PTMs (e.g. phosphorylation data)? The community also expects deep learning can help improving identification and localization of PTMs.

Authors' response:

Unfortunately in this manuscript we did not focus much on the evaluation of PTMs. For the immunopeptidomics datasets in this study, our spectral libraries and prediction models included two variable modifications of HLA peptides: M(Oxidation) and N-terminal (Acetylation). Currently

PEAKS DIA supports three common variable modifications, including N-terminal (Acetylation), M(Oxidation), and NQ(Deamidation). It may be difficult to configure any number of PTMs like in the case of DDA data. This is a common limitation for DIA analysis tools, because the libraries and the prediction models depend on the input data that were used to build them. It is difficult to generalize the models so that they are able to predict for new types of PTMs that they haven't seen in the training data, as different types of PTMs tend to have different spectrum and RT distributions. On the other hand, PEAKS Online DDA can handle any number of fixed and variable PTMs. In fact, the computing power provided by PEAKS Online shall be tremendously beneficial for PTM search. We hope to provide a comprehensive assessment of PEAKS Online performance on PTMs in a future study. We have added these limitations in the Discussion, line 290.

Reviewer #3 - question #5:

Please explain 'two-species library method' in Figure 1c, 1d, and Figure 2a.

Authors' response:

We apologize for this mistake. We use the random decoy approach to estimate FDR throughout our PEAKS Online platform, including DDA and DIA, spectral library search and database search.

Initially, when comparing to DIA-NN by Demichev et al. [35], we tried to follow their two-species library approach of FDR estimation for a fair comparison. However, at the end we decided to use random decoys to keep the consistency of our entire platform where the random decoy approach is used for DDA and DIA, spectral library search and database search.

The evaluation results in Figures 1c/d were done using the random decoy approach. For DIA-NN and Spectronaut, we generated random decoy libraries, converted them into Spectronaut format, and supplied them to DIA-NN and Spectronaut. The mistake in the x-axis label was because we forgot to update the plotting scripts accordingly.

The following is a brief description of the FDR estimation method using two-species spectral library extracted from Demichev et al. [35]:

“A spectral library created for the target organism (in this case, human) is augmented with spectra from peptides belonging to another organism (in this case, maize) not expected to be found in the sample (extensive filtering against the proteome of the target organism is performed to remove all peptides potentially originating from both organisms). Calls of these extra peptides are then considered false positive, allowing estimation of effective FDR and identification numbers that can be robustly compared irrespective of the differences introduced by the software-integrated FDR estimation methods.”

Reviewer #3 - question #6:

Why all peptides with 7 AA are new in Figure 3d? Is it because that the previous method excludes length 7?

Authors' response:

Sorry, this was our mistake, 7-mers should be excluded from the identification of HLA peptides. We have revised the analysis and updated Figure 3, the results are discussed in the main text, line 188.

Reviewer #3 - question #7:

In main text of 'Competing Interests Statement', it is 'The authors declare no competing interests'. It needs to be revised according to 'Editorial Policy Checklist'.

Authors' response:

Yes, this was a mistake. We have revised the statement according to the “Editorial Policy Checklist” as following:

“Lei Xin, Rui Qiao, Xin Chen, Hieu Tran, Shengying Pan, Sahar Rabinoviz, Haibo Bian, Xianliang He, Brenton Morse and Baozhen Shan are employees of Bioinformatics Solutions Inc.” (main text, line 303)

REVIEWERS' COMMENTS

Reviewer #2 (Remarks to the Author):

Dear Authors

All my remarks were properly addressed and the paper is ready for publication.

Kind regards

Reviewer #3 (Remarks to the Author):

The authors did great job to reply reviewers' concerns. There is still one concern left (Reviewer #3 - question #2). Although the authors provide Supplementary Figure S6 to show the PSM score distributions of random decoy peptides and target peptides, it is not clear if there is no bias in the design of decoy method (the portion of decoy peptides is small). Null search is one way to check this. Here are more details. The fragment ions are unchanged, while only the precursor mass of a MS/MS scan is shifted by adding a big mass (e.g. 100 Da). If using 10 ppm peptide tolerance to search the new MS/MS file, candidate peptides in +/- 10 ppm are not correct IDs. After database search or library search, the top-1 PSM is randomly from target or decoy peptides. Without filtering the search results, directly extract all top-1 PSM scores. If the decoy method is properly designed, the score distribution for target and decoy peptides will be similar (#target top-1 PSMs/#decoy top-1 PSMs will be close to 1:1).

Point-by-point response letter

We thank the Editor and the Reviewers for your constructive comments that greatly helped us to improve our manuscript. We have fully addressed the remaining concerns as detailed below.

Reviewer #3

The authors did great job to reply reviewers' concerns. There is still one concern left (Reviewer #3 - question #2). Although the authors provide Supplementary Figure S6 to show the PSM score distributions of random decoy peptides and target peptides, it is not clear if there is no bias in the design of decoy method (the portion of decoy peptides is small). Null search is one way to check this. Here are more details. The fragment ions are unchanged, while only the precursor mass of a MS/MS scan is shifted by adding a big mass (e.g. 100 Da). If using 10 ppm peptide tolerance to search the new MS/MS file, candidate peptides in +/- 10 ppm are not correct IDs. After database search or library search, the top-1 PSM is randomly from target or decoy peptides. Without filtering the search results, directly extract all top-1 PSM scores. If the decoy method is properly designed, the score distribution for target and decoy peptides will be similar (#target top-1 PSMs/#decoy top-1 PSMs will be close to 1:1).

Our response:

We have followed Reviewer #3's advice to perform the null search. The precursor masses were shifted by adding 100 Dalton and the database search was repeated with the new precursor masses. The new target and decoy PSM score distributions were presented in the new Supplementary Figure 6c. As Reviewer #3 suggested, the target score distribution was reduced and nearly identical to the decoy one because the target peptides no longer matched the precursor masses after the shifting. We have also added reference 41 that describes this target-decoy null search strategy. These results have been added in the revised main text.